# An Examination of COVID-19 Medications’ Effectiveness in Managing and Treating COVID-19 Patients: A Comparative Review

**DOI:** 10.3390/healthcare9050557

**Published:** 2021-05-10

**Authors:** Mahmoud Al-Masaeed, Mohammad Alghawanmeh, Ashraf Al-Singlawi, Rawan Alsababha, Muhammad Alqudah

**Affiliations:** 1Faculty of Health and Medicine, University of Newcastle, Callaghan 2308, Australia; muhammad.alqudah@newcastle.edu.au; 2Faculty of Pharmacy, Philadelphia University, Amman 19392, Jordan; ph.moh.alghawanmeh@gmail.com; 3Independent Scholar, Amman 11731, Jordan; alsinglawi@gmail.com; 4School of nursing and Midwifery, Western Sydney University, Sydney 2560, Australia; 19256232@student.westernsydney.edu.au

**Keywords:** SARS-CoV-2, COVID-19, medication, side-effects, efficacy

## Abstract

*Background*: The review seeks to shed light on the administered and recommended COVID-19 treatment medications through an evaluation of their efficacy. *Methods*: Data were collected from key databases, including Scopus, Medline, Google Scholar, and CINAHL. Other platforms included WHO and FDA publications. The review’s literature search was guided by the WHO solidarity clinical trials for COVID-19 scope and trial-assessment parameters. *Results*: The findings indicate that the use of antiretroviral drugs as an early treatment for COVID-19 patients has been useful. It has reduced hospital time, hastened the clinical cure period, delayed and reduced the need for mechanical and invasive ventilation, and reduced mortality rates. The use of vitamins, minerals, and supplements has been linked to increased immunity and thus offering the body a fighting chance. Nevertheless, antibiotics do not correlate with improving patients’ wellbeing and are highly discouraged from the developed clinical trials. *Conclusions*: The review demonstrates the need for additional clinical trials with a randomized, extensive sample base and over a more extended period to examine the potential side effects of the medications administered. Critically, the findings underscore the need for vaccination as the only viable medication to limit the SARS-CoV-2 virus spread.

## 1. Introduction

The World Health Organization (WHO) was formally notified of a severe pneumonia illness in Wuhan, China, on 31 December 2019. As of 5 January 2020, there were 59 officially tested and confirmed cases, but there were still no fatalities. However, the Chinese government enacted measures to control the virus spread through lockdowns and social gathering restrictions. This is due to the complex network of international flights and the ability of individuals to spread the illness asymptomatically [1,2,3]. The virus’s viral sequencing was separated on 7 January and its genome was shared by 12 January 2020. This was identified as the novel coronavirus SARS-CoV-2. It is commonly referred to as the COVID-19 disease. SARS-CoV-2 is not the first strain of Coronavirus that has infected humans in the past [4,5]. Other previous respiratory infections causing viruses under the same coronavirus category include the NL63 and 229E linked to bats and OC43 and HKU1 originating from rodents, all of which are endemic. Other non-endemic coronavirus strains are SARS-CoV, first recognized in November 2002. It was responsible for severe acute respiratory syndrome (SARS). The second non-endemic type was the Middle Eastern syndrome (MERS), first identified in Saudi Arabia in 2012. It remains prevalent, with over 2494 infections notified to the WHO, with 858 proving fatal [6,7] 

SARS-CoV-2 virus has been linked to six different strains. However, genetic structure analysis indicates minimal variability. In Europe and especially in Italy, the strain G was the most dominant with severe symptoms and a high fatality rate, unlike the strain L that was initially diagnosed in Wuhan, China, whose prevalence has been on the decline. The original strain was L and was followed by strain S at the beginning of December 2019. Subsequent strains have been the V and G, with G being the most globally spread strain [6,8]. The strain mutated into GR and GH strains towards February 2020. The G strain and its related mutations of GH and GR are the most widespread, accounting for 74% of all the gene mutations analyzed. The strains explain the difference in severity and symptoms exhibited by COVID-19 patients. In contrast, the minimal variance demonstrates the treatment protocols and medications’ applicability on COVID-19 patients, regardless of the strains [9,10]

As an evolving phenomenon, the global healthcare community is actively developing, treating, and currently rolling out different vaccines. This is in addition to providing treatment protocols and medication to COVID-19 patients. The treatment and medications administered range from basic nutritional recommendations for asymptomatic and mild symptoms to advanced, hospital-based medication for moderate and severe COVID-19 cases [11,12]. One of the emerging challenges is the high misinformation rates on the treatment, medication, and handling of the COVID-19 pandemic. The world has experienced a rise in fake news and misleading COVID-19 pandemic information. There is a need to develop reviews that focus on accurate and reliable COVID-19 treatment and medication data. This need informed the review development. An analysis of the existing studies indicates initial studies developed on the topic of medication efficacy in the initial stages. However, one of the challenges in the initial reviews was the lack of conclusive data. A majority examined the ongoing clinical trials and thus lacked the advantage of analyzing some of the already now-complete clinical trials. This review develops on the previous reviews through analyzing the complete and new ongoing clinical trials on the subject. The review findings demonstrate the changing context and medications efficacy in managing COVID-19 as mutant strains and variants emerge globally.

### Aim

An evaluation of the currently adopted treatment protocols and medications administered to COVID-19 patients and their effectiveness.

## 2. Materials and Methods

The review adopted a secondary literature and data search. The data and materials collection process’s core aspect was ensuring the accuracy and reliability of the findings and articles used for analysis. The search and categorization of the medication administered to COVID-19 patients were derived from the WHO solidarity clinical trials for COVID-19 treatments. In a bid to aid in its guidelines and recommendations, WHO rolled out a global clinical trial enlisting over 500 hospitals in over 30 countries and with over 12,000 patients included in the clinical trials. The clinical trials identified three treatment protocols, including antiretroviral drugs, antibiotics, and vitamins and supplements [13,14]. The search criteria and words were based on the listed and rolled out clinical trials to determine the treatment protocols’ effectiveness and recorded side effects in different treatment protocols [15]. The analysis relied on peer-reviewed databases, including Scopus, Medline, CINAHL, and Google Scholar. The peer-reviewed databases ensured the reliability and accuracy of the obtained articles. Besides, to allow for statistical and emerging data and information on COVID-19 treatment and medication, it also included studies and reports by recognized global health institutions such as WHO and the FDA. The additional data were purely hedged on obtaining reports that were a form of the clinical outcomes on the use and adoption of different COVID-19 treatment protocols. The search process was guided by a set of terms and key phrases. They included the following: COVID-19 and medication or treatment and clinical outcomes. Other synonyms used in the search process for medication included drug and patient care. These were used as extenders of the search process, especially in databases where the term medication did not yield enough searched articles.

The review’s search scope was purely limited to articles dated after 2020, with any reports developed before 2020 excluded as they did not directly relate to addressing the SARS-CoV-2 virus. The WHO solidarity clinical trials for COVID-19 treatment had the medication-effect based on the dimensions of mortality, the need for assisted ventilation, and hospital stay duration. The rationale for the dimensions used in the review was to ensure that only studies that were based on the WHO guidelines on proper and efficient COVID-19 care were included. The WHO solidarity clinical trials enumerated the scope and measures through which treatment protocol, drug, or medication regime effectiveness in handling COVID-19 would be analyzed. This informed the study basis for analyzing published trials on the different used medications and drugs. Hence, only studies focusing on these effectiveness dimensions were included in the analysis. The exclusion criteria included all articles and reviews that did not apply scientific models and designs in analyzing and comparing COVID-19 treatment and medication protocols’ effectiveness. Articles without a full PDF and lacking an English publication version were equally excluded from the study analysis. This was to ensure that only articles whose entire publication details were available were included in the study to allow for accuracy and reliability. Further, duplications of the articles across the databases were detected and all duplicates were eliminated. The obtained articles’ quality was assessed through the GRADE model. The dimensions of assessing the articles included internal consistency, perceived accuracy, and bias risk [16,17]. The ranking ranged from low to high. Only articles with a minimum-moderate rating on the GRADE scale were included in the article review analysis [18,19]. The inclusion of only those with a moderate quality assessment ensured that the included articles were relevant and enabled the review to address its aim and objectives. The elimination of some of the articles socially based on bias ensured that only articles and reviews with an objective scientific foundation in their analysis were included. This helped in weeding out some of the many articles developed hedged on myths, beliefs, and misinformation on the treatment and medication offered to COVID-19 patients. The EQUATOR PRISMA model was used to ensure that all the required and standard steps in developing the ILR review were followed and adhered to [19,20]. The key expected outcomes in the review were an understanding of the different and most commonly used medications and drugs used in managing and treating COVID-19 patients [20,21]. This is in addition to understanding, from completed and ongoing clinical trials, the efficacy of the medications on severity, hospital stay, and mortality rates among patients. The review analyzes the findings thematically based on each of the adopted medications and, under each, an examination of the different studies and clinical trials on its efficacy [22,23]. The developed review is a comparative analysis of the different medications. Thus, it does not on its own develop a statistical analysis of the medications. Instead, it relies on the comparison and contrasting of data obtained from ongoing and completed clinical trials on the analyzed drugs and medications.

## 3. Results

The review’s article search established 247 clinical trials on the topic. The review applied the exclusion of the duplications across the databases and official websites such as on the WHO and FDA websites. Two of the authors were involved in the articles’ screening, extraction, and analysis. A risk of differences in the screening process was resolved through a consensus by both authors involved. In the data extraction and screening process, it was imperative to eliminate the risk of biases. This was achieved in the review through the use of the extraction of study outcome data in a duplicates model. This was through using more than one author in the extraction process. For all the review articles and outcomes that were subjective, the analysis used two authors: one an expert in methodology and research design and a second one an expert in infectious disease management and control. This helped in ensuring that any risk of bias was eliminated as the two had to agree through consensus on any variances in the extraction and article-grading process. The extraction process was executed through the use of a data-extraction form designed and agreed upon by four of the review authors. The form stipulated the focus areas in the articles’ screening and extraction process [24,25].

In total, the review had 28 feasible clinical trials that it relied upon in developing its findings and analysis. The results indicated that in managing and treating COVID-19, the treatment protocols could be categorized into three main clusters: (1) the use of antiretroviral, (2) the use of antibacterial, and (3) use of multivitamins and food supplements. A summary of the findings and the exclusion and inclusion criteria is as illustrated in the PRISMA Figure 1. The findings are illustrated below.

### 3.1. Antiretroviral

#### 3.1.1. Veklury (Remdesivir)

Statistics and studies indicate that the Veklury (Remdesivir) drug was proposed to be used as an antiviral drug in managing and treating patients with COVID-19. The recommendations were based on earlier tests and clinical randomized trials developed in relation to drug use in treating and managing Ebola patients [26,27]. Tests on humans and animals demonstrated its safety and effectiveness in fighting the Coronavirus family viruses. The SARS-CoV-2 family falls within this category, making the drug use reliable. This was later approved by regulatory bodies such as FDA. However, its regulation and authorization have been restricted to treating adults over the age of 12 years and people weighing at least 40 kgs (88 pounds). The major condition for its global approval has been its use restriction to a pure hospital environment. Moreover, its use has been highly discouraged outside the said environment in the care for COVID-19 patients. The veracity and the value of the drug in managing COVID-19 patients are hedged on randomized controlled clinical trials on patients using the medications. Overall, it has been linked to improved well-being for mild and severe symptoms [28,29]. One randomized study was developed by the National Institute of Allergy and Infectious diseases. Using a randomized sample base of 1062 patients, 541 patients received Veklury (Remdesivir) and placebo (521) and tandardized care (14). The recovery index was hedged on either being discharged from the hospitals or remaining in the hospital, not needing oxygenation or any other medical treatment protocol. The trial’s findings indicated a median recovery of 10 days for those receiving Remdesivir compared to 15 days for the placebo group. The improvements at 15 days were equally higher for the Remdesivir-treated patients [30].

An equal, randomized trial was developed through an open-label, multicenter trial comprised of 60 trial sites and 13 subsites in the United States (45 sites), Denmark (8), the United Kingdom (5), Greece (4), Germany (3), Korea (2), Mexico (2), Spain (2), Japan (1), and Singapore (1). The findings indicated that the sample base receiving Remdesivir at day 5 had higher improvements and recovery rates at day 11 of infection than those receiving standardized care [31]. A similar study using the same design and test days established related findings. It further established statistical differences in mortality differences, with those receiving standardized care having a higher mortality rate. Additional studies developed in England indicate that Remdesivir has an effect on lowering the mortality rates among COVID-19 patients. The trial established that the Kaplan–Mier estimates of mortality at day 15 were 6.7% for Remdesivir and 11.9% for placebo, while at day 29, they were 11.4% for Remdesivir and 15.2% for placebo [32]. Additional studies have equally supported the lower mortality rates for patients using Remdesivir with severe COVID-19 complications [32,33].

Despite the improvements and value established in managing and treating COVID-19 patients, studies have demonstrated some of the side effects of using Remdesivir. They include swelling around the eyes, lips, and under the skin, allergic reactions resulting in changes such as blood pressure and heart-rate rhythm effects, and suspected injury to the liver due to demonstrated increased liver enzymes [34,35].

#### 3.1.2. Oseltamivir

As a retroviral drug, its effect and success in managing and binding with previous coronavirus family strain in the past resulted in its consideration in treating and managing COVID-19 [36,37]. Studies and clinical trials have been developed examining the drug’s integrity and effectiveness in handling COVID-19. One such test is a simulated model using the Swiss-Model. It constructs the N-terminal RNA binding domain (NRBD) of the nucleoprotein (NC) papain-like the protease (PLpro) and the RNA-directed RNA-polymerase (RdRp) of the severe acute respiratory syndrome coronavirus (SAR-CoV-2). The proteins are aligned to assess the drug’s ability to bind the virus. Findings indicated that PLPrO and RdRp were structurally similar to the influenza A neuraminidase, with TM-scores of 0.30077, 0.19254, 0.28766, 0.30666, and 0.34047 [38].

Further, the active center of the 3CL pro was similar to the active center of the neuraminidase of influenza A. The carboxylic acid for Oseltamivir was favorable in binding the active site of 3CL pro. However, its inhibitory effect was minimal as compared to the control groups. Besides the simulated experiments, randomized clinical trials were developed examining its actual impact and effectiveness in managing COVID-19 patients. The studies indicate a correlation between Oseltamivir use in early treatment and the mitigation of the patients’ symptoms. A clinical randomized trial illustrated that the use of Oseltamivir in early treatment (ET) initiated within 24 h lowered the duration of the fever (31 ± 21 h) as compared to those receiving the drug in late treatment (LT- 24 h after the fever start) where the fever lasted longer (94 ± 38 h). The study illustrated that the use of Oseltamivir reduced the fever severity and time taken to recover for COVID-19 patients if administered as early treatment [38]. Further studies such as those from Akram, Azhar, Shahzad, Latif, and Khan [39] and Rosa and Santos [40] demonstrated similar effectiveness in early treatment. Akram, Azhar, Shahzad, Latif, and Khan’s study was a clinical trial examining the effects of the tested drugs, including Hydroxychloroquine Phosphate/Sulfate vs. Oseltamivir vs. Azithromycin, each alone and in combination in seven control groups. The impact on COVID-19 patients was measured on day seven through a clinical recovery metric. The findings indicated that the Oseltamivir was effective when used in early treatment, but its effect and impact on reducing coronavirus nucleic acid from the throat declined as patients crossed the first 24 h of infection. If used in early treatment, it resulted in significant clinical improvements, including lowered fever levels by day seven of the trial compared to the standardized care group. However, its effects as a treatment protocol after the early stages were ruled out as less effective. Additional findings indicated a minimal relationship between the use of Oseltamivir and the patients’ length of stay in the hospitals and its impact on the need for invasive ventilation, renal replacement therapy, and ECMO within 14 days of hospitalization [40]. The aggregate finding was that Oseltamivir effectively managed symptoms and fever in the early stages of infection but was less effective in offering treatment for severe and advanced COVID-19 patients’ conditions.

#### 3.1.3. Favipiravir

The orally administered Favipiravir (FabiFlu) inhibits the RNA polymerase (RdRP). This slows down the virus replication rate. It is estimated that if the inhibitors function at full capacity in inhibiting all the viruses, they would stop or, at the least, derail the virus replication cycle. This allows the body enough time to develop antibodies to fight the virus, thus helping patients recover [41,42]. A randomized clinical trial on the drug was conducted on 150 patients. It established a value of the antiretroviral use and standardized care compared to the offering of standard care alone. By administering 3600 mg on day one (1800 mg, twice a day) and 1600 mg (800mg twice daily) for the consecutive days up to 14 days, the use of Favipiravir enhanced faster time to clinical cure and delayed the need for oxygen therapy for patients [43,44]. The median to clinical cure time in using the drug was reduced by 2.5 days, with oxygenation need delayed by over a week.

Clinical studies on Favipiravir have been developed globally in India, Russia, China, and Japan, all showing positive and promising results. This has triggered more clinical trials on drug use in nations such as the USA and the UK. Studies in China and Russia indicate a direct relationship between the use of the drug and reduced viral load and the virus clearance’s median days’ reduction compared to standard treatment. The findings demonstrate the effects of the drug in reducing the clinical treatment period for patients with COVID-19. However, the results also assert that its effectiveness is higher if administered earlier in the infection cycle as compared to the patients who receive the drug dosage late in their infection cycle [44,45]. It was linked to reduced hospitalization period, reduced mortality, and reduced and delayed need for patients’ oxygenation reliance if administered early enough.

### 3.2. Anti-Bacterial

In managing and treating COVID-19, it is imperative to understand that it is a viral disease. Thus, it cannot be prevented or treated with anti-bacterial use. According to WHO usage guidelines, such drugs are only effective in mitigating bacterial infections. However, anti-bacterial drugs such as Fluoroquinolones, Levofloxacin, and Moxifloxacin have commonly been used on COVID-19 patients. Their use has been based on the understanding that with the viral infection under COVID-19, the patients’ immune is suppressed [46,47]. This exposes them to the risk of co-infection by bacterial infections. One of the drugs under experimentation for treating COVID-19 patients’ co-infection is Azithromycin. The drug has anti-inflammatory effects. With COVID-19 moderate and severe cases, there are often inflammation symptoms resulting from the body’s overactive immune response. The use of the Azithromycin drug has an effect on reducing the severity of the inflammation symptoms among patients. The application of the Azithromycin drug on SARS-CoV-2 virus is based on its effectiveness in simulated controlled experiments on other related viral infections, such as Ebola and Zika [48,49].

Further, a non-randomized clinical trial in France indicated that the use of Azithromycin and Hydroxychloroquine resulted in reduced severity of respiratory tract infection, reduced hospitalization due to the elimination of bacterial co-infection, and a reduction in the viral carriage. When used together, Azithromycin added to the hydroxychloroquine’s effectiveness in managing co-bacterial infections. The study was non-randomized and was a small clinical trial, making its efficacy for generalization and in concluding a challenge [47,48,49]

Adopting anti-bacterial drugs such as hydroxychloroquine and chloroquine has not been linked to any improvement. In a recovery trial, patients using hydroxychloroquine had a 27% mortality rate compared to the control group’s 25% at 28 days of medication use. Further, the patients using hydroxychloroquine were registered as having a higher need for mechanical ventilation than the control groups [50]. The findings under the WHO’s solidarity trial indicated that hydroxychloroquine was in no way associated with improvement in mortality compared to other groups. This was in addition to the clinical trial on outpatient and non-hospitalized COVID-19 patients denoting no correlation between its use and reduction in symptoms. Thus, anti-bacterial use has been ruled out as a treatment medication for COVID-19. However, their use should only be limited to COVID-19 patients with bacterial co-infections, such as on their respiratory system [44,45]. The side effects noted by healthcare professionals are the risk of increased anti-bacterial resistance using anti-bacterial in managing and treating COVID-19 patients.

### 3.3. Multi-Vitamins and Supplements

The third category in managing and treating COVID-19 recommended and discussed by healthcare professionals is using multi-vitamins and supplements. The vitamins’ and minerals’ inclusion as an alternative in managing COVID-19 patients is based on their inherent oxidation and anti-inflammatory effects. A majority of severe COVID-19 patients require invasive ventilation or mechanical ventilation to ensure the right level of oxidation. This is due to the lung damage done by the viral infection [47,48]. The use of vitamins plays a key role in promoting oxidation, thus lowering the risks and the probable need for invasive ventilation among patients. Further, studies demonstrate that the use of vitamins and minerals reduces the risk of inflammation, thus reducing the symptoms related to lung inflammation and general inflammation due to sensitive high immune response. Vitamins C and D boost the body’s immunity [51,52,53]. This is linked to either reducing the risk of infection severity or shedding off the virus, thus infecting others. Trials have linked the level of infection severity to increased viral load and the risk of shedding off the virus to others. Through vitamin and supplement use, the boosting of immunity reduces the severity of the infection and, by extension, reduces the risk of mortality rates among COVID-19 patients. Unfortunately, an analysis of existing clinical trials indicates that most of the trials are in progress [51,52,53]. This means that there is a lack of a definitive, scientific, evidence-based model to establish the relationship between the use of multivitamins, supplements, and minerals on COVID-19 infection severity, symptoms, and mortality rates. This finding is expected to emerge with the completion of the progressing clinical trials.

A summary table for the findings is illustrated in Table 1.

## 4. Discussion

A critical analysis of the findings indicates that the obtained findings address the overall review aim and objectives. The review aimed at examining the different types of medications/drugs used in caring for and managing COVID-19 patients. This is in addition to an examination of the drugs’ efficacy in terms of the patients’ hospital stay, mortality rates, and severity of the illness. The findings demonstrate that the use of antiretrovirals such as Remdesivir, Oseltamivir, and Favipiravir has increased efficiency in lowering the viral load and replication rates. The slowed-down replication of the virus offers the body enough time to fight and respond to the virus. Additionally, the use of multivitamins and supplements allows for the boosting of the body’s immunity, while anti-bacterial use and value are limited to addressing any bacterial infections that may result from COVID-19-related complications.

Although related to other coronavirus family viruses, the virus’s genome structure is unique and different. In its genome structure, the SARS-CoV-2 resembles the wild bat virus more than the previous coronavirus strains, such as MERS-CoV. The spike protein requires six amino acids, and SARS-CoV-2 only shares one of these with SARS-CoV [54,55]. Further, SARS-CoV-2 has a unique subunit of the spike protein that determines the viral infectivity and the host range. This is linked to a virus mutation increasing the virus multiplication rate. The SARS-CoV-2 virus gene structure differs significantly from previous coronaviruses. This makes use of the past antiretroviral drugs that worked on previous types of coronavirus a challenge. An examination of antiretroviral drugs such as Remdesivir, Oseltamivir, and Favipiravir indicates their limitations in managing and controlling the virus multiplication cycle [56,57,58]. For instance, although it matches the virus’s inner active nucleus structure, the Remdesivir drug is unable to control the replication cycle. This is because its inhibitory properties are weak and lack enough strength to entirely bind the virus. Hence, although they slow down replication and multiplication, antiviral drugs lack the complete inhibition to end the multiplication cycle fully.

Further, drug effectiveness is linked to the viral load and infection rate among patients. Drugs such as Oseltamivir are effective when the viral load and the cycle are at their formative stages (within 24 h of infection). The effectiveness of inhibiting declines as the virus cycle advances with prolonged infection time. This has led to the general agreement among professionals that antiviral drugs are effective as an early treatment protocol for patients. Besides, the use of drugs does not constitute curing. Instead, it only slows down the infection rate, allowing the body’s immune system enough time to resist and fight the virus [54,55,56,57,58,59]. Drugs are a support army in fighting the COVID-19 virus. They inhibit cell multiplication that damages essential organs such as the lungs. Consequently, the need for mechanical and invasive ventilation and oxygenation is significantly reduced. Further, through slowing down the viral replication, the viral count remains low, allowing for a shorter clinical cure period than patients receiving standard treatment [59].

The use of antibiotics and vitamins with supplements is not a treatment protocol for COVID-19. The findings indicate that two medication forms are not stipulated under the recommended and standard treatment protocols for patients with COVID-19. Antibiotics are linked to health risks such as antibody resistance and have no direct impact on slowing down or inhibiting virus cycle and multiplication. Nevertheless, one of the impacts of COVID-19 is suppressed immunity. Suppressed immunity and hospitalization expose patients to the risk of co-infections, mainly with bacterial infections. The studies recommend the limitation of the use of antibiotics to such cases as treating secondary infections [60,61]. At the onset of the pandemic, misinformation on the virus and treatment led to instances where patients in home-based care bought antibiotics off the shelf and used them as part of their medication [59]. This has been linked to potential long-term complications, drug resistance, and no proven benefits in managing COVID-19. Recommendations are strong on terminating any form of antibacterial use as a medication to treat primary COVID-19 symptoms [44,45]. Equally, the use of multivitamins, although with the previous linkage to anti-inflammatory effects and acting as oxidants, remains unreported in clinical trials on its impacts in slowing down infection rates and hastening the recovery process. Vitamins are recommended for their impact on the body’s immunity boost.

The review’s main limitation is the lack of large clinical trials on medications’ side effects. The review is based on studies developed in 2020, and as such, the time frame for the clinical trials was limited. A majority of clinical trials had either a relative sample base or the process was sped up to allow recommendations and decisions amidst the COVID-19 pandemic. Thus, the review’s analyzed findings fail to demonstrate some of the long-term implications and side effects of the recommended and used medications. The side effects could manifest in the long run, with some emerging within a year from the start of the clinical trials. Aspects such as the long-term complications on the lungs and liver among COVID-19 patients are slowly emerging long after their treatment and clinical cure of COVID-19. The review was unable to highlight the long-term side effects, and future studies can focus on the emerging and manifesting side effects in the long-run period.

The review’s implications are the compilation of reliable and up-to-date data on clinical trials, clinical outcomes, and the extent of the use of different medication and treatment protocols in treating COVID-19 patients [50,55]. Misinformation on the medications (especially the effectiveness and impact on the mortality stay in hospitals, clinical cure time, and the mean recovery and reliance on mechanical and invasive ventilation) has led to limitations with some healthcare providers and patients resisting the medications. The analysis and findings serve as a basis for evaluating the scientific efficacy and the short-term impact on COVID-19 and its long-term implications on patients’ health and well-being. The findings demonstrate that although the antiviral drugs recommended by WHO have side effects, they have a value and shorten the clinical cure time, reduce the severity of the illness, and reduce and delay the need for invasive and mechanical ventilation [51,54]. However, their effectiveness is mainly at the early treatment stage. Further, the findings affirm the need for vitamins, minerals, and supplements to boost patients’ immunity. The results affirm why the WHO rejected and discouraged the use of antibiotics as a treatment medication due to the lack of a direct value to COVID-19 patients and the long-term limitations, such as antibiotics resistance.

## 5. Limitations

The initial review search was done in the year 2020, and thus the study scope had a limitation of articles published in 2020. However, consideration for extending the review search into subsequent years was provided for in future publication versions of the review. Additionally, the review had a limitation in the reliance on the use of only articles published in the selected databases. Any literature not available on the peer-reviewed databases was excluded. This could have exposed the review to the risk of excluding and thus failing to consider critical and useful literature in the findings development.

## 6. Conclusions

The review analysis has demonstrated that antibiotics are not recommended as a treatment and are highly inappropriate due to the risk of long-term challenges. Furthermore, the review findings emphasize the need to develop a SARS-CoV-2 vaccine. This is based on the realization that the current medication and treatment protocols are insufficient to handle and manage COVID-19. Vulnerable categories such as those with pre-existing conditions and the elderly remain at high mortality risk. The medications proposed are insufficient to support vulnerable categories. The development of a vaccine remains the feasible medical practice in addressing the SARS-CoV-2 virus spread and the mortality risks.

## Figures and Tables

**Figure 1 healthcare-09-00557-f001:**
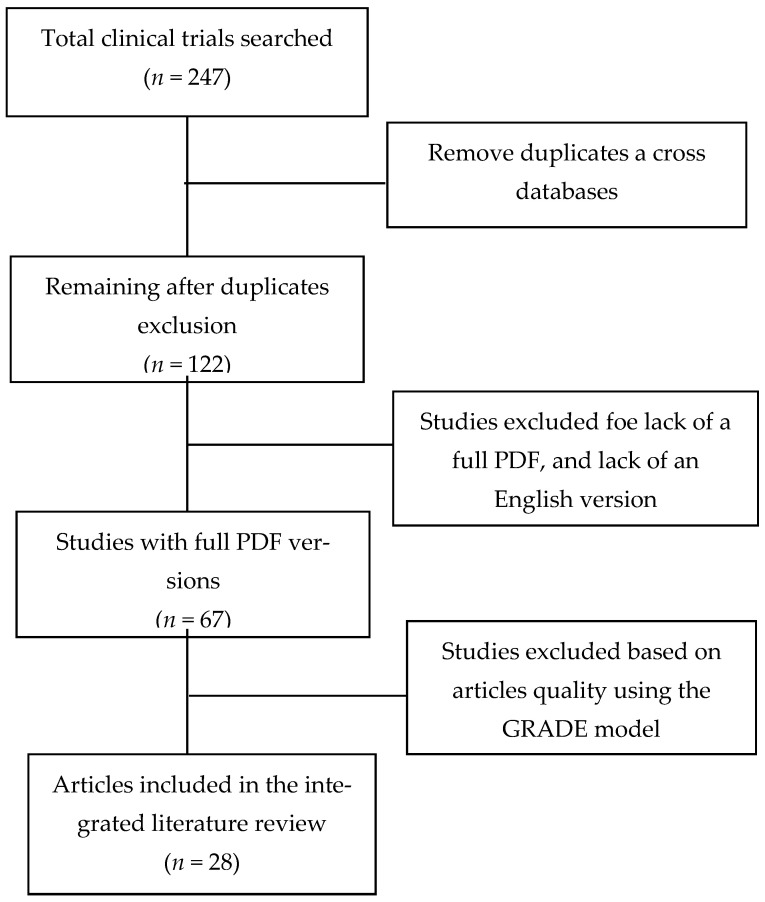
Flowchart of the study selection process.

**Table 1 healthcare-09-00557-t001:** Findings Summary.

Treatment Option/Protocol	Key Findings
Anti-retroviral (Veklury (Remdesivir), Oseltamivir, Favipiravir)	Used to slow down the viral replication and mutation Helps in slowing down the viral multiplication and viral load Offers The body enough time to respond and use its immunity to fight off the virus infection
Anti-biotics	Not directly used to manage the virus replication and mutation Used as a secondary medication for secondary bacterial infections resulting from a Covid-19 infection
Multivitamins and Supplements	Not directly helping in lowering the virus mutation and replication. Critical in helping boast the body immunity thus reducing the severity of the viral infection

## Data Availability

All the sued reviews, literature and data in the review is sourced from peer reviewed databases namely Scopus, Medline, Google Scholar, and CINAHL. Additional articles are sourced from the WHO and FDA websites.

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
