# Peer review of "An Examination of COVID-19 Medications’ Effectiveness in Managing and Treating COVID-19 Patients: A Comparative Review"

_healthcare, 2021, doi:10.3390/healthcare9050557_

Round 1

Reviewer 1 Report

  1. reference numbers should be placed in square brackets [ ]; https://www.mdpi.com/journal/healthcare/instructions#references.
  2. In the initial search process, why the word “drug” had not been used along with "Covid-19' and 'medication' or 'treatment' and 'clinical outcomes.'
  3. And, still the article search was purely limited to 2020, I understand that this article might have drafted in 2020 and I feel It would be adding more value if the author plan to extend the search to 2021 also. sometime year of the accepted article may change after the issue released.
  4. Again, studies focusing on mortality, the need for assisted ventilation, and hospital stay duration only focused on the review? Why?   
  5. Article without pdf and lacking an English publication were equally excluded from the study analysis.  
  6. “exclusion of the duplications across the databases” good point to highlight here and its indeed needed.
  7. Check similar review “ Overview of the First 6 Months of Clinical Trials for COVID-19 Pharmacotherapy: The Most Studied Drugs” and justify the significance of your review.
  8. NIH-COVID-19 Treatment Guidelines frequently updating with recent information (https://www.covid19treatmentguidelines.nih.gov/whats-new/), author need to discuss the novelty in this review.
  9.  

Author Response

Respected Prof / Dr

Hope you are doing well

I appreciate the comments. The comments majored mainly in the background section on rationale and the methods and materials section. I appreciate that you pointed out some of the areas and aspects I had either omitted or had not given enough weight. I have added these details including synonyms used in the search process, the inclusion and exclusion criteria, and the rationale of the review. I have also revised the references both in text and on the reference list as required. All changes have been tracked for ease of reference and verification.

Have a wonderful time

Sincerely

Mahmoud

Reviewer 2 Report

This is a timely review to examine the effectiveness of Covid-19 medications in managing and treating Covid-19 patients but there are issues of scientific soundness of describing methods and presenting results. I find a lack of clarification about screening of literature, review process, data analysis, outcome measures, and reporting results including those involved in the review steps.

My comments:

  1. According to author statement, “The obtained articles' quality was assessed through the GRADE model. The dimensions of assessing the articles included internal consistency, perceived accuracy, and bias risk. The ranking ranged from low to high. Only articles with a minimum moderate rating on the GRADE scale were included 105 in the article review analysis (Kowalczyk, Pleszczynska, and Ruland, 2004; Al-Masaeed, 106 Al-Soud, Alkhlaifat and Alsababha, 2020).”

It is not clear whether authors used any risk of bias assessment tools to categories studies into low, medium and high-quality study? It is also unclear how author used GRADE model before risk of bias assessment? Did authors use any specific risk of bias assessment tool and why has no statement. There are no citations supporting GRADE model and risk of bias assessment tools? I struggle to find any tables presenting risk of bias assessment results and GRADE guided results as well. This question the quality of the review. If those have been performed need to describe in the manuscript and present relevant results.

  1. There is no clear statement about the specific outcomes that authors are looking at in the method section. How these outcomes have been measured and how data were analysed have no description
  2. There is no information in the methods about which statistical analysis or any other descriptive analysis that was performed to measure outcomes or compare results.
  3. Does author use any standard tool like PRISMA for conducting the review? I can’t find any PRISMA flow chart demonstrating article selection as well.
  4. How many authors were involved in the article screening, data extraction and data analysis process and who were they is unclear?
  5. Does author use any validated data extraction tool is not clear
  6. Authors describe results using a set of relevant studies for a particular medicine but it could be presented in tables clearly showing trial method, study context, patient samples, interventions, and outcomes including effectiveness. This would increase clarity, and readers understandability as well.
  7. In the discussion section, key findings can be presented in the first paragraph in line with research questions, aims and objectives of this review rather than repeating background.
  8. In discussion, some paragraphs are just the repetition of result section but can be improved with comparison with international literature. 

Author Response

Respected Prof / Dr

Hope you are doing well

I appreciate the comments and feedback offered. I am sorry that some of the finer details required were omitted such as the number of authors responsible for the screening and extraction process and the bias risk assessment models and tool used. I have included these details in the revised manuscript as they were adopted in the review process.  I have also improved my discussion and results sections as requested by the reviewer. I hope the improvements make the review better and easier to understand. All changes in the document have been tracked for ease of verification of the changes made.

Sincerely

Mahmoud

Round 2

Reviewer 1 Report

This article may be considered for further process

Author Response

Respected Reviewer

Thanks for the feedback. 

Reviewer 2 Report

This is improved version but following points can to be considered to improve the quality of this paper

1. References that have been used to support the use of GRADE seem incorrect.

2. No references have been used for the risk of bias tool and PRISMA guidelines

3. A table describing study and results needs more comprehensive information about the trials to understand findings

4. PRISMA diagram is needed with comprehensive details

5. Description needs international literature for the comparison of findings

6. Limitation of the study can be more specific to the review process along with the limitation of using the findings

Author Response

Respected Reviewer

Thanks for the feedback. This is how I addressed the comments

  1. On references for TRADE model- Added references 60, 61 page 3
  2. References for PRISMA too Added references 58, 59 page 3
  3. Findings on a table Added a table- Added a table on a different file
  4. Detailed PRISMA diagram- Added a PRISMA figure in a different file attached
  5. Using international literature for international findings- Added references 51, 54, 56 page 7, 8
  6. Limitations of the study- Added a study limitations section.

Have a wonderful time
